# Infantile Hemangioma: A Cross-Sectional Observational Study

**DOI:** 10.3390/life13091868

**Published:** 2023-09-04

**Authors:** Florica Sandru, Alina Turenschi, Andreea Teodora Constantin, Alexandru Dinulescu, Andreea-Maria Radu, Ioana Rosca

**Affiliations:** 1Dermatology Department, “Elias” University Emergency Hospital, 011461 Bucharest, Romania; florica.sandru@umfcd.ro; 2Department of Dermatovenerology, University of Medicine and Pharmacy “Carol Davila”, 020021 Bucharest, Romania; 3Pediatric Hospital Ploiesti, 100326 Ploiesti, Romania; alinaburcuta@yahoo.com; 4Pediatrics Department, National Institute for Mother and Child Health “Alessandrescu-Rusescu”, 20382 Bucharest, Romania; andreea.constantin@drd.umfcd.ro; 5Department of Pediatrics, University of Medicine and Pharmacy “Carol Davila”, 020021 Bucharest, Romania; 6Department of Pediatrics, Grigore Alexandrescu Emergency Hospital for Children, 011743 Bucharest, Romania; alexandru.dinulescu@rez.umfcd.ro; 7Department of Neonatology, University of Medicine and Pharmacy “Carol Davila”, 020021 Bucharest, Romania; ioana.rosca@umfcd.ro; 8Neonatology Department, Clinical Hospital of Obstetrics and Gynecology “Prof. Dr. P. Sârbu”, 060251 Bucharest, Romania

**Keywords:** infantile hemangioma, premature infants, very low birth weight

## Abstract

(1) Background: With an incidence of 4–10%, infantile hemangiomas (IH) are the most encountered benign tumors in infancy. Low birth weight (LBW), prematurity, female sex, multiple gestations, and family history of IH are some of the statistically proven risk factors for developing IH. The aim of our study was to evaluate the prevalence of IH in our clinic and its connection to maternal and perinatal factors. (2) Methods: We conducted a cross-sectional study, over three years (2020–2022), at the Clinical Hospital of Obstetrics and Gynecology, “Prof. Dr. P. Sârbu”, in Bucharest, Romania. (3) Results: During this period, 12,206 newborns were born and we identified 14 infants with infantile hemangioma. In our study, the prevalence of infantile hemangioma was 0.11%. The prevalence of IH in pregnancies obtained through in vitro fertilization was 1%, in twin pregnancies it was 2.27%, and in those with placenta previa, it was 4.16%. (4) Conclusions: Our findings provide a solid image of the prevalence of IH in our country and underline that the development of IH is strongly connected to maternal and perinatal variables, such as: preterm newborns, in vitro fertilization, high blood pressure, anemia, hypothyroidism, placenta previa, and twin pregnancy.

## 1. Introduction

With an incidence of 4–10%, infantile hemangiomas (IH) are the most encountered benign tumors in infancy [1,2]. They are characterized by a phase of rapid growth, followed by a progressive involution. This feature is what sets them apart from vascular malformations [3,4,5]. A third can be observed at birth, 40% emerge during the next 4–6 weeks, and the remaining third have developed by the age of 6 months [6,7]. The proliferation phase is usually completed by 9 months old according to the literature, succeeded by a gradual downsize that can last up to 48 months old [8].

Low birth weight (LBW), prematurity, female sex, Caucasian race, progesterone treatments, multiple gestations, and family history of IH are some of the statistically proven risk factors for developing IH. However, in the literature, there have been more risk factors proposed, such as in vitro fertilization (IVF), older maternal age, placental abnormalities, and maternal smoking [9,10].

In terms of clinical appearance, they can be superficial, presenting as a unique pink-red macule, papule, or plaque, with the papillary dermis being the limit of their extension or, in the case of deep infantile hemangiomas, they appear as a pale blue, subcutaneous nodule. They can arise anywhere, external or on internal organs (liver, central nervous system); however, the most common localizations are the skin and soft tissue [6,11]. Around 80% of IHs develop on the face and neck, with the anterior cheek, forehead, and pre-auricular region being the most encountered [6,12].

IH with limited or no proliferation constitutes a unique variant of IH known as IH with minimal or absent growth, of which, in contrast to normal IH, roughly 75% of these vascular lesions are visible at birth. During the proliferative phase, only a small portion of the lesion is affected, usually involving the periphery. Their involution phase resembles the other IHs [13,14]. However, in terms of clinical appearance, IH with minimal or absent growth has some distinguishable particularities: a pink macule with areas of vasoconstriction and a blanching-appearing peripheral halo, accompanied by fine telangiectasia [13,14].

Although still not exactly deciphered, the pathogenesis is considered to be multifactorial. One of the key players in developing infantile hemangioma is local hypoxia, which results in the inhibition of the hypoxia-inducible factor-pathway (HIF-pathway) and chaotic growth. Another factor to be considered is the capability of IH cells to differentiate in perivascular cells, contributing to the proliferative phase [15,16].

In the majority of cases, the diagnosis of IH is based on clinical appearance. However, there are certain investigations that may contribute to the diagnosis stage, such as: dermoscopy, high-frequency ultrasonography, or biopsy [10]. In dermoscopy, IH are seen as well-demarcated lacunae, round or oval-shaped, with a color ranging from red to reddish-blue, accompanied by isolated vessels on a variably colored background. Dermoscopy may also facilitate the differential diagnosis between thrombosed hemangioma, which appears as a clearly demarcated jet-black zone accompanied by the typical lacunae, and melanoma [17].

In terms of treatment, up until 2008, the recommended agents were topical, intralesional, or systemic corticosteroids. However, their adverse effects, such as increased infection risk, skin atrophy, osteoporosis, hyperglycemia, hypertension, and growth suppression hampered the course of treatment [18,19,20,21]. Since Leaute-Labreze et al. published their statement paper on propranolol and its benefic effects on IH, several studies have been conducted regarding β-blocker safety administration in infants and, at this point in time, β-blockers represent the first-line therapy in IH [22]. Local therapy with topical beta-blockers such as propranolol or timolol in a thin layer, two or three times a day until the age of 12 months old, is indicated in cases of small and superficial hemangiomas [23]. In cases of large IHs or those that pose a risk of complications, oral propranolol is the first line of therapy administered at a dose of 1–3 mg/kg/day until 12 months of age. There are several notable side effects of β-blockers, such as the exacerbation of psoriasis, the onset of psoriatic arthropathic erythema multiforme, alopecia, Peyronie’s disease, or lichenoid reactions, thus a certain amount of care is needed when prescribing this medication [24,25]. Another option of treatment could be surgery or pulsed dye laser; however, β-blockers remain the primary treatment [26,27].

The aim of our study was to analyze the prevalence of neonates with cavernous hemangioma in our clinic and to compare our data with those obtained from the literature. Furthermore, we evaluated the connections between the prevalence of IH and its known risk factors, such as: preterm newborns, in vitro fertilization, high blood pressure, anemia, hypothyroidism, placenta previa, and twin pregnancy. We want to underline the importance of interdisciplinary collaboration between neonatology and pediatric dermatology through a faster detection of cases and efficient management, highlighted by a good long-term patient prognosis. As our clinic is a third-level maternity hospital, it takes a special interest in premature follow-up.

## 2. Materials and Methods

We conducted a cross-sectional observational study investigating neonates with cavernous hemangioma who were admitted to our clinic for a period of over 3 years (January 2020–December 2022). The study cohort included 14 patients with cavernous hemangioma. The inclusion criteria were newborns with cavernous hemangioma, while the exclusion criteria were incomplete data, neonates with other benign skin disorders or rashes, and no workup or intervention. The follow-up period was 2 years. The diagnosis of IH was made based on clinical appearance and dermoscopy examination. The gender of the newborns, the way the pregnancy was obtained (naturally or in vitro fertilization), complications in pregnancy (obesity, hypertension, infections, anemia, cervical cerclage, hypothyroidism, ureterohydronephrosis, placenta previa), family history of IH, the weight at birth, gestational age at birth, delivery method (vaginal birth or cesarean section), and Apgar score were recorded for each patient. Among these 14 patients, only 1 patient received treatment for IH. According to the World Health Organization (WHO) guidelines, anemia in pregnancy is defined by maternal hemoglobin levels below 110 g/L during the first and third trimester and below 105 g/L in the second trimester. Preeclampsia was defined by systolic blood pressure of 140 mmHg or greater or diastolic blood pressure of 90 or greater on two measurements at least 4 h apart or systolic blood pressure of 160 mmHg or more or diastolic blood pressure of 110 mmHg or more in a shorter period of time, after 20 weeks of gestation [28]. Data were collected from the hospital’s electronic register using simple random sampling to reduce bias. Data collection was carried out using Microsoft Excel. Prematurity refers to a gestational age below 37 weeks, with subcategories of late preterm (32–37 weeks of gestation), very preterm (28–30 weeks), and extremely preterm (<28 weeks) [29]. We used the WHO classification to classify the newborns by the birthweight: 1501–<2500 g-Low birth weight (LBW), 1001–1500 g-Very low birth weight (VLBW), and ≤1000 g Extremely low birth weight (ELBW) [30]. SPSS version 26 was used to analyze the data. We used Fisher’s Exact Test to compare the prevalence of hemangioma in those naturally obtained vs. IVF and in term vs. preterm newborns.

## 3. Results

After neonatal screening in the maternity ward, over the 3-year study period, 12,206 newborns were born, of which 1250 newborns were hospitalized in the intensive care unit; 191 (1.56%) pregnancies of the total newborns (12,206) were obtained through in vitro fertilization; 1144 (9.37%) were preterm according to the case definition; we identified 14 infants with cavernous hemangioma. In our study, the prevalence of cavernous hemangioma was 0.11%. The mean age of the mothers at the time of birth was 35.07 ± 6.12 years, the minimum age being 25 years old and the maximum being 44 years old. The characteristics of the study group are presented in Table 1 and the distribution of complications in pregnancy is presented in Figure 1 (Table 1, Figure 1).

The prevalence of hemangiomas in the naturally obtained pregnancies was 0.1%, and in those obtained through IVF, it was 1%. We compared the prevalence using Fisher’s Exact Test and it was statistically significant (*p* = 0.02) (Table 2).

The prevalence of cavernous hemangioma is higher in preterm newborns (1.13%) than in those born at term (0.0001%) and, according to the Fisher’s Exact Test, is statistically significant (*p* < 0.001) (Table 3).

According to the Fisher’s Exact Test, we found no correlation between hemangiomas and obesity (*p* = 0.149) (Table 4).

The correlation between high blood pressure and increased prevalence of hemangiomas is statistically significant (*p* = 0.003) (Table 5).

Anemia in pregnancy correlates with a lower prevalence of newborns hemangioma (*p* = 0.001) (Table 6).

No correlation was found between hypothyroidism and hemangiomas (*p* = 0.284) (Table 7).

Those with placenta previa had statistically significant higher prevalence of hemangiomas (*p* = 0.001) (Table 8).

There was no correlation between hemangiomas and the mode of delivery (*p* = 0.1) (Table 9).

Hemangiomas had a higher prevalence in twin pregnancy compared to single pregnancy (*p* = 0.004) (Table 10).

## 4. Discussion

The prevalence of IH varies in the literature. Munden A et al. reported a prevalence rate of 4.5% in their study [3]. However, we found a 0.11% prevalence rate in our study, which is significantly lower. A contributing factor to this low rate could be the small study group of patients.

There are several risk factors that have been linked to developing IH, such as: female gender, low birth weight, abnormalities of the placenta, vaginal bleeding during early pregnancy, use of in vitro fertilization, and family history of hemangioma [3,31]. The study conducted by Munden A et al. included 29 infants with IH; 47% of them were male, while 53% of them were female [3]. However, in our study, there were more male patients with hemangioma identified (57.14% vs. 42.85%). In terms of birth weight, we found the majority of infants were born with very low birth weight or extremely low birth weight (35.71% and 57.14%, respectively), which is consistent with the findings in the literature. Drolet et al. published a study that included 420 infants with IH; 79.8% of them weighed under 2500 g at birth [32]. In addition, Drolet et al. proved that low birth weight was the most notable risk factor, leading to a 40% risk increase in IH for every 500 mg drop in weight [32]. Moreover, Munden A et al. also found that out of 29 infants with IH, 14.8% were born prematurely, 14.3% were extremely premature, and only 3.9% of them were term infants [3]. However, a recent study conducted on a Chinese population did not identify preterm birth and LBW as risk factors for developing IH [33]. Another large-scale nationwide epidemiological study carried out on 85,244 Japanese mothers did not find any statistically significant association between LBW and IH, and there was no statistically significant difference in the risk of IH between pre-term, post-term, and full-term deliveries [31]. However, both studies that showed no connection between IH and LBW or premature newborns were only conducted on Asian populations, therefore, this could be attributable in part to racial and ethnic variances. In our study, both these features were found to be a predisposing factor of IH, according to the Fisher’s Exact Test, which was statistically significant.

Another instance in which LBW is encountered is preeclampsia. Owing to the fact that there have been multiple studies that have demonstrated the fact that preeclampsia is a risk factor for IH, we raise the question as to whether there is a relationship between hypertension and IH [33]. According to the results in our study, hypertension during pregnancy is associated with a higher prevalence of newborns with IH.

The invasion of extravillous trophoblasts plays a crucial role in establishing the attachment of the placenta to the uterus and facilitating nutrition acquisition for the development of the fetus over the whole duration of pregnancy [34]. Because the thyroid hormone plays a significant role in facilitating this process, the concentrations of thyroid hormone or thyroid-stimulating hormone (TSH) have the potential to impact the onset and progression of preeclampsia [33]. Thus, we evaluated the correlation between the prevalence of IH and hypothyroidism. However, we did not find a statistically significant connection.

Since IH and the placenta exhibit a comparable life cycle, which consists of a phase of cell proliferation, then followed by a time of morphological stability, and ultimately leads to involution, North et al. claimed that IH may originate from the presence of ectopic placental tissue [3,35]. Both IH and placentas exhibit the expression of certain cell surface markers, including GLUT1, Lewis Y antigenic FcyR11, and merosin [35]. These proteins are often not seen in tissues other than those derived from neuronal or placental origins [35]. Barnes et al. later conducted transcriptome clustering analysis on placenta and vascular tumors, revealing that the microarray expression patterns of IH and placental tissue had striking similarities. This finding provides evidence for a potential association between the two [36]. Gutierrez et al. conducted an analysis of placentas obtained from a sample of 26 newborns with a weight less than 1500 g, whereby half of the infants had IH. Pathological alterations were seen in the placentas of all pregnancies associated with IH, whereas only 23% of the healthy newborns had placental abnormalities [37]. Moreover, in the study by Munden et al., placenta previa was found in 6.9% of cases [3]. The association between IH with placenta abnormalities is also consistent with our study results, as the prevalence of placenta previa among newborns with IH is higher than in those without IH (4.16% vs. 0.09%), with this result being statistically significant (*p* = 0.001).

As Dickison P et al. found in their study, IVF is yet another important factor in predisposing newborns to developing IH, as infants conceived through IVF were presumed more likely to have IH than those naturally obtained (25% vs. 6%) [1]. This is consistent with our study, as the prevalence of hemangiomas in the naturally obtained pregnancies was 0.1%, and in those obtained through IVF, it was 1%.

Genetic predisposition in IH is controversial. However, 21% of patients from the study by Dickison et al. had a family history of IH [11]. In another study conducted over a period of 3 years, out of 185 patients, a third of them had a positive family history of IH and, in the majority of cases, it was a first-degree relative [38]. In our study, we did not identify any patient with a positive family history of IH.

Multiple-gestation pregnancy is recognized as a risk factor for IH. However, for a long period of time, there was a lack of clarity as to whether several factors contribute to this predisposition, such as a lower gestational age, LBW, uterine crowding, maternal high blood pressure, genetic predisposition, or placental insufficiency. In a prospective study that included 201 pairs of twins, 37% of the pairs were concordant for the presence of IH. Moreover, the study provided confirmation that the etiology of IH is influenced by several factors. Multiple predisposing variables may synergistically interact, leading to the attainment of a threshold necessary for the manifestation of clinical symptoms. This research provides evidence in favor of the notion that lower gestational age, LBW, and the feminine biological sex are risk factors for IH, and these variables may have a greater impact than the susceptibility caused by a single gene [39]. In our study, the prevalence of IH was higher in twin pregnancies compared to single pregnancies (2.27% vs. 0.09%) and it was statistically significant (*p* = 0.004).

The hypoxia theory may explain some of the risk factors for developing IH (e.g., anemia); many other conditions, such as malignant tumors feature hypoxia via cell proliferation and angiogenesis [40,41,42]. The upregulation of hypoxia-inducible factor-1α (HIF-1α) is a pivotal factor in the cellular responses elicited by hypoxia. The early-life exposure to hypoxic conditions results in an increased production of HIF-1α, therefore facilitating the restructuring of the vascular system into a complex network of a branched vascular tree. This adaptive response is essential for meeting the augmented demands for oxygen and nutrients [33,43]. Anemia is frequently associated with pregnancy and has a multifactorial etiology. Anemia during pregnancy is often associated with a reduction in hemoglobin levels, which leaves the fetus more vulnerable to hypoxia [33]. This condition has been shown to be associated with increased incidences of maternal and perinatal morbidities, including preeclampsia, placenta previa, preterm delivery, and LBW [44]. As anticipated, pregnant rats with iron-deficient anemia exhibited a noteworthy rise in HIF-1α levels when compared to the control group [45]. A matched case-control study conducted between 2017–2020, on 1033 IH patients, proved that anemia during pregnancy should be considered a key independent risk factor for IH [33]. However, in our study, anemia in pregnancy proved to be correlated with a lower prevalence of hemangiomas in newborns, according to the Fisher’s Exact Test. This result may be explained by the small number of patients included in our study.

In our study, we also evaluated if there is a connection between the prevalence of IH and obesity, as well as the mode of delivery. However, we did not find any statistically significant correlation between either of these factors, nor did we find any studies in the literature with which to compare this data.

Despite their benign feature, infantile hemangiomas can put the patient at risk for developing several complications, such as ulceration, functional impairment, or physical disfigurement. With a prevalence of 10–15% among patients, ulceration is the most encountered complication of IH in a proliferative phase. It mainly occurs in large IH or the ones situated on the lips, neck, or in the ano-genital region. Due to the potential of irreversible destruction to the involved anatomic structures, unremitting ulceration should enforce aggressive treatment [46]. However, the patients included in our study had no complications during the follow-up period.

In some cases, large cutaneous IHs can be part of a syndrome, such as PHACE. The acronym stands for posterior fossa malformation, hemangiomas, arterial, cardiac, and eye abnormalities, and has been associated with maternal preeclampsia or placental defects [47]. Haagstom A.N et al. reported an incidence of 31% of PHACE among infants with large hemangioma on the neck (>22 cm^2^) [48]. In a prospective study that aimed to investigate the prevalence of PHACE syndrome, out of 1906 children enrolled, only 2.3% met the diagnosis criteria for this syndrome [49].

Another syndrome that manifests with the presence of IH is LUMBAR syndrome, its acronym standing for: lower body IH, urogenital anomalies, ulceration, myelopathy, bony deformity, anorectal and arterial malformations, and renal anomalies [10,50,51,52]. In the case of this syndrome, IH is usually encountered on the lower part of the body, with the most frequent localization being the lumbosacral and perineum area, sometimes extending to one lower limb. Other cutaneous manifestations include lipomas, skin tags, and atrophic macules [10,50,53]. However, none of the patients enrolled in our study had either PHACE syndrome or LUMBAR syndrome.

The study by Leaute-Labreze et al. published in 2008 was a turning point in IH management, since two newborns with IH underwent systemic therapy with propranolol for cardiac pathology and, during the course of treatment, a remarkable involution of IH was observed [22]. Henceforward, propranolol, which is a non-selective β-blocker, became the preferable choice in the treatment of IH. At a dosage of 1 mg/kg per day, propranolol delivers notable results in newborns with IH [54]. This was also the case for one of our patients included in the study, for whom propranolol was commenced for 10 months, followed by 2 months of local ointment. After this course of treatment, the patient’s IH was considerably ameliorated.

A prospective multicenter, randomized clinical trial, which included 377 patients, analyzed the efficacy and safety of both propranolol and atenolol among patients with IH [55]. At the initial evaluation, after 6 months, the propranolol group had a better response rate than the atenolol group (93.7% vs. 92.5%). However, after two years, both groups presented a similar percentage of response to treatment. Moreover, adverse effects were more frequently encountered in the propranolol group [55]. The results from this clinical trial show that atenolol is a safe option for problematic IH, either as the first-line agent or as an alternative for patients who have contraindications or intolerance to propranolol [55]. Another clinical trial that compared nadolol and propranolol in terms of efficacy and safety showed that the patients who received nadolol obtained a faster and better response in treating IH. Although nadolol had more favorable outcomes in this clinical trial and it poses a lower risk of crossing the blood-brain barrier, further research is needed to certify its superiority over propranolol [56]. Our patient was treated with oral propranolol for 10 months, followed by 2 months of local ointment, with a significant clinical amelioration of the vascular lesion. No adverse effects were encountered during the course of treatment.

Another treatment option for patients with non-satisfactory results after propranolol therapy is Nd:YAG 1064 nm laser. In an open-label prospective cohort study, 30 patients were included and treated with the aforementioned laser for their residual IH after propranolol treatment [57]. Out of thirty patients, eighteen had a great response, ten had a good response, and only two had a moderate response. The most encountered side effects were erythema, edema, and minimal pain, with no important adverse reaction reported. Therefore, the authors concluded that Nd:YAG 1064 nm laser therapy should be considered as a second-line treatment for residual IH [57].

The main limitation of our study is the small cohort of patients, especially when referring to treatment, since we only had one patient that received therapy. However, to our knowledge, there have not been many studies that have analyzed the prevalence of IH published in our country and the correlation between IH and its risk factors, and that could be considered one of the strengths of our study.

## 5. Conclusions

In summary, we found a rather low prevalence rate of IH in our clinic compared to the data in the literature. Furthermore, our study underlined that the development of IH is strongly connected to maternal and perinatal variables, such as preterm newborns, in vitro fertilization, high blood pressure, anemia, hypothyroidism, placenta previa, and twin pregnancy. Moreover, in terms of treatment, we did not encounter any side effects from propranolol therapy.

Our findings provide a solid image of the prevalence of IH in our country and the associated maternal and perinatal risk factors. Patients who have these risk factors should undergo closer monitoring and follow-up in the clinic so that early diagnosis can take place and treatment options may be made to enhance outcomes.

## Figures and Tables

**Figure 1 life-13-01868-f001:**
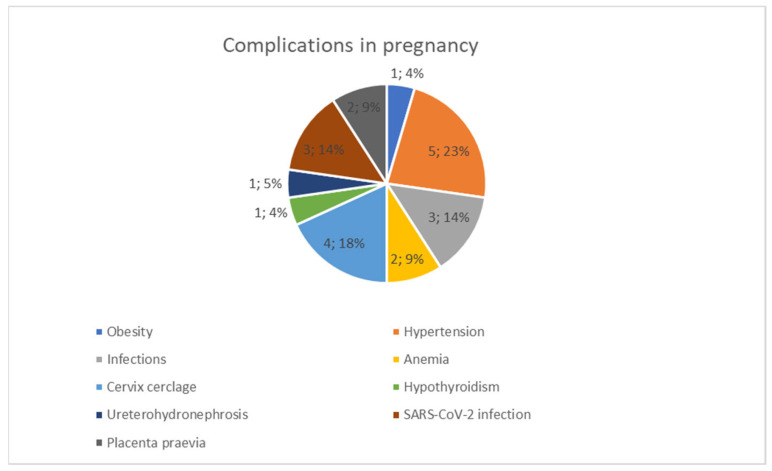
Complications in pregnancy.

**Table 1 life-13-01868-t001:** Characteristics of the study group (*n* = 14 neonates).

Variables	Results
Gender of newborn babies	
Male	8 (57.14%)
Female	6 (42.85%)
Pregnancy obtained	
Naturally	12 (85.71%)
In vitro fertilization	2 (14.28%)
The birth	
Our maternity	13 (92.85%)
Other maternity	1 (7.14%)
Complications in pregnancy	
Obesity	1 (7.14%)
Hypertension	5 (35.71%)
Infections	3 (21.42%)
Anemia	2 (14.28%)
Cervix cerclage	4 (28.57%)
Hypothyroidism	1 (7.14%)
Ureterohydronephrosis	1 (7.14%)
SARS-CoV-2 infection	3 (21.42%)
Placenta praevia	2 (14.28%)
Perinatal	
Rupture of membranes > 48 h	2 (14.28%)
Cervix culture—Streptococcus group B	2 (14.28%)
Urinary tract infection	1 (7.14%)
Family History of Hemangioma	-
Gestational Age	
>38 weeks	1 (7.14%)
34–36 weeks	-
28–30 weeks	5 (35.71%)
<28 weeks	8 (57.14%)
Weight at birth	
>2500 g	1 (7.14%)
1501–<2500 g (LBW **)	-
1001–1500 g (VLBW ***)	5 (35.71%)
≤1000 g (ELBW ****)	8 (57.14%)
IUGR *	-
Twins	2 (14.28%)
Delivery	
Cesarean section	13 (92.85%)
Vaginal birth	1 (7.14%)
Apgar score	
8–10	1 (7.14%)
6–7	6 (42.85%)
<5	7 (50%)
Postnatal	
Transfusion	13 (92.85%)
Mechanical ventilation	10 (71.42%)
Antibiotics	13 (92.85%)

* intrauterine growth restriction, ** low birth weight, *** very low birth weight, **** extremely low birth weight.

**Table 2 life-13-01868-t002:** Prevalence of hemangiomas in natural pregnancy and in vitro fertilization (IVF).

	Natural Pregnancy	In Vitro Fertilization	*p* *
Total	12,015	191	0.02
Cavernous Hemangioma	12 (0.1%)	8 (1%)	

* Fisher’s Exact Test.

**Table 3 life-13-01868-t003:** Prevalence of hemangiomas in preterm vs. term newborns.

	Term Newborns	Preterm Newborns	*p* *
Total	11,062	1144	<0.001
Cavernous Hemangioma	13 (0.001%)	1 (1.13%)	

* Fisher’s Exact Test.

**Table 4 life-13-01868-t004:** Prevalence of hemangiomas in newborns from obese vs. normal weight mothers.

	Obesity	Normal Weight	*p* *
Total	2583	9623	0.149
Cavernous Hemangioma	1 (0.03%)	13 (0.13%)	

* Fisher’s Exact Test.

**Table 5 life-13-01868-t005:** Prevalence of hemangiomas in newborns from mothers with high blood pressure vs. normal blood pressure.

	HBP	NBP	*p* *
Total	976	11,230	0.003
Cavernous Hemangioma	5 (0.51%)	9 (0.08%)	

* Fisher’s Exact Test.

**Table 6 life-13-01868-t006:** Prevalence of hemangiomas in newborns from mothers with anemia vs. mothers with normal levels of hemoglobin.

	Anemia	Normal Hb	*p* *
Total	6884	5322	0.001
Cavernous Hemangioma	2 (0.02%)	12 (0.22%)	

* Fisher’s Exact Test.

**Table 7 life-13-01868-t007:** Prevalence of hemangiomas in newborns from mothers with hypothyroidism vs. mothers with euthyroidism.

	Hypothyroidism	Euthyroidism	*p* *
Total	244	11,962	0.284
Cavernous Hemangioma	1 (0.4%)	13 (0.1%)	

* Fisher’s Exact Test.

**Table 8 life-13-01868-t008:** Prevalence of hemangiomas in newborns from mothers with placenta previa vs. mothers with normal placenta.

	Placenta Praevia	Normal Placenta	*p* *
Total	48	12,158	0.001
Cavernous Hemangioma	2 (4.16%)	12 (0.09%)	

* Fisher’s Exact Test.

**Table 9 life-13-01868-t009:** Prevalence of hemangiomas in C-section delivery vs. natural birth.

	C-Section	Natural Birth	*p* *
Total	9341	2865	0.1
Cavernous Hemangioma	13 (0.13%)	1 (0.03%)	

* Fisher’s Exact Test.

**Table 10 life-13-01868-t010:** Prevalence of hemangiomas in twin pregnancy vs. single pregnancy.

	Twin Pregnancy	Single Pregnancy	*p* *
Total	88	12,118	0.004
Cavernous Hemangioma	2 (2.27%)	12 (0.09%)	

* Fisher’s Exact Test.

## Data Availability

The data is available from the corresponding author upon reasonable request.

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
