# Peer review of "Infantile Hemangioma: A Cross-Sectional Observational Study"

_life, 2023, doi:10.3390/life13091868_

Round 1

Reviewer 1 Report

Essentially a case report, the study does look at their institution for rates of occurrence of infantile hemangioma and risk factors for such occurrence. Those data are interesting, and add to our understanding of this uncommon condition. 

there are some editorial tweaks that could strengthen the message.

Reviewer 2 Report

Dear authors,  

Although I find your manuscript very interesting and the chosen topic is an important one in pediatric pathology, I have some observations and recommendations. 

The abstract needs to be reorganized: too much general information in the Introduction and too little in the Discussions and Conclusions chapter.
Introduction: Part of the information presented in the Introduction can be transferred to the Discussions chapter; your study is an original study and not a literature review. Also, the aim of the study should be the last paragraph of the Introduction section, not the first of the Methods section.

Results: -Line 184-193: this paragraph is redundant; you presented the data in the Table 1. The same for Figure 1.  

Since the hemangioma is usually not present at birth but appears later, it is important to mention until what age your patients were followed. If the follow-up period is short, the low prevalence reported by your study in relation to the specialized literature could be explained in this way. 

A case presentation, even a special one, has no place in a cross-sectional study. You should choose: either focus on the original study, or do a case presentation with a literature review.

Discussion: Here you should discuss your results in relation to the data from the specialized literature and explain the differences. Also, a section on Limitation of the study is mandatory. 

What is the conclusion of your study?

Round 2

Reviewer 2 Report

Dear authors,

I found your manuscript improved.